# Leprosy in an Adopted Woman Diagnosed by Molecular Tools: A Case Report from a Non-Endemic Area

**DOI:** 10.3390/pathogens12020165

**Published:** 2023-01-20

**Authors:** Anna Beltrame, Maria Concetta Fargnoli, Charlotte Avanzi, Laura Sollima, Elena Pomari, Antonio Mori, Silvia Stefania Longoni, Lucia Moro, Pierantonio Orza, Mary Jackson, Francesca Perandin

**Affiliations:** 1Department of Infectious—Tropical Diseases and Microbiology, I.R.C.C.S. Sacro Cuore Don Calabria Hospital, Negrar di Valpolicella, 37024 Verona, Italy; 2College of Public Health, University of South Florida, Tampa, FL 33612, USA; 3Dermatology Unit, Ospedale San Salvatore, 67100 L’Aquila, Italy; 4Department of Biotechnological and Applied Clinical Sciences, University of L’Aquila, 67100 L’Aquila, Italy; 5Mycobacteria Research Laboratories, Department of Microbiology, Immunology and Pathology, Colorado State University, Fort Collins, CO 80523, USA; 6Pathology Unit, Ospedale San Salvatore, 67100 L’Aquila, Italy

**Keywords:** leprosy, *Mycobacterium leprae*, adopted, diagnosis, PCR

## Abstract

Coupled with its rarity in non-endemic areas, the clinical heterogeneity of leprosy makes diagnosis very challenging. We report a diagnosis of multibacillary leprosy in a 22-year-old Indian woman, adopted at the age of 10 and living in Italy. The patient presented with painful skin lesions on the face, trunk, and lower and upper extremities, associated with dysesthesia and a motor deficit in her left leg following corticosteroid therapy interruption. Histopathology results from the skin lesions suggested leprosy, but no acid-fast bacilli were identified. Molecular biology in a center specializing in tropical diseases confirmed the diagnosis, allowing prompt and adequate treatment. Genotype analysis allowed the identification of a genotype 1D of *M. leprae*, facilitating the epidemiological investigation of the plausible infection origin. No resistances to rifampicin, dapsone, or ofloxacin were detected. Leprosy will continue to exist in high-income nations, and the incidence may rise over time due to increasing migration and globalization. CARE guidelines were followed.

## 1. Introduction

Also referred to as Hansen’s disease, leprosy is a neglected, chronic, and infectious disease caused by *Mycobacterium leprae* and *M. lepromatosis* [1,2]. Although its prevalence has decreased by almost 95% since the introduction of multidrug therapy (MDT) in 1981, the disease has yet to be eradicated [3]. In 2020, 127,396 new cases of leprosy were reported in 127 countries, including 74% in India, Brazil, and Indonesia [4]. Leprosy in Europe has become rare as of the second half of the 20th century, and the most recent cases were primarily imported from endemic regions [5,6]. Up to the 1980s, Italy only had autochthonous leprosy cases in distinct, endemic areas (Northern Tuscany, Eastern Sicily, Calabria, Puglia, and Liguria) [7]. With the increase in migratory flows from low-income countries, the proportion of imported cases has progressively increased [5,8,9]. Since 2003, cases in the native Italian population have only been documented among those spending significant time in endemic regions, such as expatriates and missionaries [5,9]. The challenging nature of the leprosy diagnosis, along with the lack of medical readiness only reinforced by the COVID-19 pandemic, is most likely the cause of the disparity between the reported number of cases and the current estimates [5]. 

*M. leprae* and *M. lepromatosis* are spread primarily through tiny droplets (aerosol) from the nose and mouth of subjects with untreated leprosy during close and frequent contact [10]. A minority of subjects infected with the mycobacterium develop the disease after an incubation period of 2–10 years, with a broad spectrum of clinical manifestations mainly affecting the skin, peripheral nerves, mucosal surfaces of the upper respiratory tract, and eyes, depending on the patient’s immune system [1]. In the presence of a strong and organized specific cellular response, the disease manifests itself in localized, tuberculoid form (tuberculoid leprosy). In contrast, in the absence of a specific immune response, the bacilli proliferate and disseminate in the lepromatous form (lepromatous leprosy) [11]. Most patients exhibit intermediate forms of the disease (borderline-tuberculoid, mid-borderline, and borderline-lepromatous leprosy). All of these forms could be detected through skin lesion histology [11]. However, 4–8% of leprosy patients can have the disease confined to peripheral nerves (primarily ulnar, median, posterior tibial, and peroneal nerves) without skin involvement, which is defined as primary neural leprosy, and requires the identification of acid-fast bacilli (AFB) on nerve histology [12]. Used as a confirmatory test, the Ridley–Jopling histopathology classification requires invasive diagnostic techniques, and its sensitivity varies on the disease stage [12]. According to the latest WHO guidelines for the management of leprosy, a straightforward diagnosis can be made with at least one of the following: (i) definite loss of sensation in a pale (hypo-pigmented) or reddish skin patch; (ii) a thickened or enlarged peripheral nerve with a loss of sensation and/or weakness of the muscles supplied by that nerve; (iii) the presence of AFB in a slit-skin smear (SSS) [13]. Leprosy is also classified according to the number of skin lesions with the paucibacillary (PB) form (five or less skin lesions) and the multibacillary (MB) form (more than five skin lesions), or the identification of AFB in an SSS [13]. This classification is important, as MB means the patient is contagious, and family members must be screened to reduce the risk of transmission to the community [14,15]. The WHO guidelines do not recommend additional tests to diagnose leprosy [13]. However, the limitations of standard diagnostic tests, the risk of patient stigmatization following an uncertain diagnosis, and the fear of recommending empiric MDT with possible adverse effects could delay diagnosis, the management of patients and their close contacts, and early treatment. 

The present case highlights the challenges encountered in confirming a leprosy diagnosis in an adopted Indian woman living in Italy. We place particular emphasis on how enhanced diagnostic techniques used in a dedicated tropical disease center validated the diagnosis of leprosy and the start of treatment.

## 2. Case Report

A 22-year-old woman presented to the Dermatology Unit of the San Salvatore Hospital in L’Aquila, Italy, in 2021 with a two-month history of painful skin lesions appearing initially on the lower extremities and then expanding to the face, trunk, and upper extremities. Notably, she reported dysesthesia and a motor deficit in her left leg. Her medical history showed she had been affected by myasthenia gravis associated with a thymoma. The thymoma was removed in 2015, and the myasthenia gravis was treated with pyridostigmine 60 mg four times a day and prednisone 50 mg every day. The prednisone was progressively reduced and then suspended after 5 years, approximately 10 months before the onset of the skin lesions. Dermatological examination revealed multiple, annular and linear, erythematous macules and papules on the upper limbs, lumbar region, and, more extensively on the lower limbs (Figure 1). Their dimensions ranged from 0.5 cm to 4 cm, with edges ill-defined without infiltrating, with a dry surface, and with the presence of hypoesthesia. Additionally, a round, erythematous-violaceous plaque with a diameter of 2 cm, well-defined edges, and an anergic smooth surface was evident on the forehead. The eyes, ears, and nose were normal. The peripheral nerves of both the hands and feet were not thickened. Neurological examination revealed tactile hypoesthesia and thermo/pain in the lesions, extending to the entire left forefoot and calf along the midline of the back of the lower left leg and where calf muscles join to the Achilles’s tendon, corresponding to the sural nerve areas. Electroneuromyography showed a sensory axonal neuropathy of the left sural nerve. Routine laboratory tests were unremarkable. A histological examination performed on two skin biopsy specimens on the right arm showed non-necrotizing dermo-hypodermal epithelioid granulomas with an intense lymphoplasmacytic infiltrate, deeply involving the adnexal structures and nerve branches (Figure 2). After Ziehl–Neelsen (ZN) staining, AFB were not found. The patient’s records state that she was adopted while in India at the age of 10 and has lived in Abruzzo, Italy ever since.

Upon the clinical and pathological suspicion of leprosy, the patient was referred to the IRCCS Sacro Cuore Don Calabria Hospital, Negrar di Valpolicella, Italy, for further investigation. The diagnostic tools used (microscopical, and serological methods) are summarized in the Appendix A. AFB were evident upon re-evaluation of the histological samples, and *M. leprae* antibodies against the specific phenolic glycolipid-I (PGL-I) were positive on serum (6.37, positive if ≥1) [16]. Microscopic analysis of a nasal swab and multiple SSS were negative for AFB. However, the search for genomic sequences of *M. leprae* in real-time PCR in DNA extracted from skin biopsies confirmed the presence of the bacterium (Table 1) and the leprosy diagnosis [17,18]. 

The extraction of DNA was performed in Italy, and the PCR was performed double-blind in Italy and the USA. Total DNA was extracted from paraffin-embedded skin samples (n = 10 sections, each of 5 microns) by the MagCore Automated Nucleic Acid Extractor (RBC Bioscience, Seoul, Korea), using the 401 MagCore Genomic DNA Tissue Kit as per the manufacturer’s instructions. DNA was eluted in a final volume of 50 μL and stored at −20 °C. The primers and hydrolysis probe used in this study were described in Truman and were designed to target the *M. leprae*-specific repetitive element RLEP [17]. The analysis was performed with the Applied Biosystems 7500 Fast dx Real Time PCR platform (Thermo Fisher Scientific, Waltham, MA, USA). The 25 μL total reaction volume contained 12.5 μL of Sso Advanced Universal Probe Supermix 2× (Bio-Rad, Hercules, CA, USA), 5 μL of the template DNA, 0.4 μL of non-acetylated BSA 5 mg/ml, with forward and reverse primers at a final concentration of 900 nM, and 250 nM of hydrolysis probe in DNase-free water. PCR amplifications were carried out as follows: initial activation at 95 °C for 3 min, 45 cycles of amplification for 15 s at 95 °C and 60 °C for 30 s. A high and a low dilution of synthetic specific DNA target RLEP amplicon were used as positive controls, and blank distilled water was included as negative control. Reactions were performed in duplicate. To verify that negative cases were not due to the presence of PCR inhibitors, human beta-actin gene was used as a PCR inhibition-positive endogenous control. All beta-actin gene reactions were performed using the Sso Advanced Universal Probe Supermix 2x (Bio-Rad, CA, USA) as follows: 5 μL of the same template DNA were extracted from the same samples, 100 nM of each primer, 200 nM of hydrolysis probe and 0.5 μL non-acetylated BSA mg/ml in 25 μL total reaction volume. The primers used for drug resistance screening were taken from previous studies and are described in Appendix A. For genotyping, we first checked the presence of the main *M. leprae* genotype circulating in India (genotype 1D) using the primers previously described [18] (Appendix A). For each sample, 5 μL of the starting materials, negative control (water) or positive control (*M. leprae* DNA strain Thai-53, NR19352 diluted 1:100) were used in 50 μL reactions with the QuickLoad taq 2×S Master Mix, (New England Biolab, Ipswich, MA, USA). Quality was assessed on agarose gel 1% TAE. Amplification started with a 3 min initial denaturation step at 94 °C, followed by 40 cycles of 30 s denaturation at 94 °C, 30 s annealing at 58–60 °C (all PCR primers in Appendix A with their respective annealing temperatures), and extension at 72 °C for 30 s, with final extension at 72 °C for 5 min. Amplicon sequencing was performed by Genewiz (United States), and data analysis was performed using CodonCode Aligner v9.0.2. Analysis of the *rpoB, folP,* and *gyrA* drug resistance-determining regions associated with rifampicin, dapsone, and ofloxacin resistance, respectively, excluded any drug resistance mutations associated with these genes (Appendix A). Genotype analysis revealed that the *M. leprae* strain exhibited a genotype 1D, a common genotype found in India, China, Brazil, and South Africa (Appendix A). 

A diagnosis of borderline leprosy (MB following the WHO classification) was, therefore, reached. WHO multidrug treatment was started with both rifampicin 600 mg and clofazimine 100 mg monthly, as well as both clofazimine 50 mg and dapsone 100 mg daily. The patient experienced dapsone hypersensitivity syndrome 1 week after starting treatment, which required discontinuing the dapsone. The patient remains under treatment with rifampicin and clofazimine, which have been well tolerated with a progressive resolution of clinical signs. The adoptive parents of the patient did not show any symptoms or clinical signs of leprosy, and PGL-I results were negative in a single sampling which will be repeated annually. Written informed consent was obtained from the patient to publish this case report and the accompanying images.

## 3. Discussion

To diagnose a case of leprosy, especially in non-endemic areas and in initial forms of the disease, can be a great challenge [1]. First of all, clinicians have to include leprosy in the differential diagnosis of other cutaneous and nervous lesions [1]. Moreover, the spectrum of clinical manifestations can be very broad depending on different immune systems of the patients and further complicate the recognition of the disease [1]. The diagnosis of leprosy requires skilled healthcare professionals to recognize the clinical signs and symptoms in at-risk individuals, such as those born in endemic regions and having migrated to Europe. In the United Kingdom, the clinical diagnosis of leprosy was reported to be delayed in 80% or more of patients on their initial medical visit by a median time of 1.8 years [19]. This is particularly significant because the disease can spread to the community and result in lifelong deformities and impairments if left untreated [3]. Few cases have been reported in adopted individuals [20,21]. Our patient is a young woman born in India and adopted 10 years after birth. The patient’s origins and the identification of genotype 1D *M. leprae*, which typically circulates in India, suggest the patient most likely acquired the infection in her country of birth and manifested the disease in Italy, 12 years after her arrival [22]. In fact, it is known that leprosy can have an incubation period of up to 30 years, while the triggers leading from subclinical to symptomatic manifestation are not fully understood [1]. Some medications can trigger a subclinical mycobacterial infection. Leprosy and tuberculosis are reported among patients under treatment for autoimmune dermatological, rheumatological, and gastroenterological disorders with anti-TNF alpha agents, or when these agents are stopped [23,24,25,26]. This is related to the granuloma disorganization caused by the interaction between mycobacterium, lymphocytes, epithelioid cells, and macrophages [27]. Glucocorticoids are effective at controlling inflammatory and autoimmune diseases thanks to their anti-inflammatory and immunosuppressive effects, yet they can also reactivate infections such as tuberculosis [26]. Our patient developed symptoms once the corticosteroids used for the myasthenia gravis were stopped. It is, therefore, likely that the 5 years of corticosteroid administration played a role in disease manifestation in this case. 

Pathogen detection in leprosy is difficult, as culturing leprosy bacilli is not possible in vitro. The demonstration of AFB in the skin/nerve lesions or in the SSS is the "gold standard" for diagnosis confirmation only for MB cases, as sensitivity in PB cases is very low [13]. In fact, the sensitivity of the Ridley–Jopling histological classification depends on the choice of biopsy site (often a single site), on the form of leprosy (61% and 46% of borderline-tuberculoid and borderline-lepromatous leprosy, respectively), on the staining used (the Fite stain should be preferred over ZN stain, as the latter may fail to stain many *M. leprae* given they are weakly AFB), and on the skills of the pathologists [12,28]. Further, it is very important not to rule out a diagnosis of leprosy in the absence of AFB in biopsies. Granulomatous inflammation and/or perineural inflammation are useful for suspecting leprosy and require adjunctive tests. Our patient was correctly subjected to the first level of tests (dermatological examination and skin biopsy) soon after the onset of signs and symptoms. In our case, no AFB was seen in the biopsy obtained in the first hospital, but the typical histological features led clinicians to request specialist evaluation, with additional tests before obtaining a clear diagnosis. Furthermore, the results of the SSS performed in the specialized center were negative, confirming its limits. In fact, the microscopic method requires qualified personnel to perform, process (stain), and then conduct (interpret) the microscopic analysis of smears [29]. Contrary to its high specificity, the sensitivity of SSS is rarely greater than 50% depending on the expertise of the microbiology technicians, but also on the stage of the disease [30,31]. Depending on the bacterial load, SSS is useful for the diagnosis of lepromatous, borderline lepromatous, and histoid leprosy, but less sensitive for the tuberculoid form and neuritic leprosy [12,32]. Additionally, as a minimum of 10,000 bacilli/g of tissue are required for microscopic detection, smears can be negative in PB leprosy [33]. Regarding the PGL-I test, whereas its specificity is 99%, the sensitivity ranged from 87% (77.1–97.3%) in MB patients to 33% (15.2–74.4%) in PB patients, limiting its use to screen the household contacts and follow up the treated patients [17]. Recently, new laboratory assays with increased accuracy have become available [34,35]. Nucleic acid-based testing is the most interesting, considering its high performance, low cost, and ease of implementation and performance in reference laboratories [33,36,37,38]. Current PCR-based assays to detect *M. leprae* infection in DNA extracted from nasal secretion, SSS, or biopsies are reported as highly sensitive, specific [33,36,37,38,39,40,41,42], and with greater diagnostic accuracy than microscopy, as well as better able to differentiate leprosy from other dermatological conditions, resulting in a confirmatory test [39,40,41]. The sensitivity of PCR ranges from 34 to 80% in patients with PB forms to greater than 90% in patients with MB forms of the disease [39]. Despite being unavailable commercially, this tool can be easily implemented in reference laboratories and is now available at our institution. However, it is important to highlight that, especially in patients with PB leprosy, it is possible that the results of all available diagnostic methods (microscopy, PGL-I, PCR) will be negative. 

Another important parameter to consider after the diagnosis of leprosy is the drug resistance of the strain. Drug-resistant strains are circulating globally, though mainly in India and Brazil [43,44]. The molecular identification of specific mutations in the *rpoB*, *folP1*, and *gyrA* genes is the fastest way to detect resistance to rifampicin, dapsone, and ofloxacin (a second-line drug). However, there is currently no known molecular target for clofazimine [43,44]. The mouse footpad infection is the most used animal model to measure *M. leprae* drug resistance. Nevertheless, the method requires a high number of bacilli in skin lesions and animal inoculation within 72 hours of the skin biopsy. Results are usually available 12 to 18 months after inoculation due to slow pathogen growth [10]. In our case, there is only one laboratory in Europe with the necessary expertise, and the low quantity of *M. leprae* in lesions did not allow for inoculation of mice. However, the absence of mutations in all three genes was suggestive of the sensitivity of the strain for rifampicin, dapsone, and ofloxacin, permitting rapid and adequate treatment. 

Finally, the new molecular diagnostic methods can define the genetic diversity of *M. leprae*, allowing one to clarify the origin of the infection. In our case, the genotyping defined that the pathogen was acquired in the country of birth and not in the country of adoption, where in any case leprosy was endemic in the past. Then, the identification of genotyping can be useful to describe the epidemiology of leprosy worldwide [18]. 

In Italy, leprosy is diagnosed only through clinical evaluation by a few clinical experts assisted by microscopic and histological methods, with all of the limits described above. This report describes the challenges in confirming the diagnosis of leprosy in an adult woman from Italy born in a highly endemic country for leprosy and how the use of molecular biology implemented in a center specializing in tropical diseases allowed the diagnosis confirmation, the prompt and adequate treatment and care as well as epidemiological investigation to determine the origin of infection.

## Figures and Tables

**Figure 1 pathogens-12-00165-f001:**
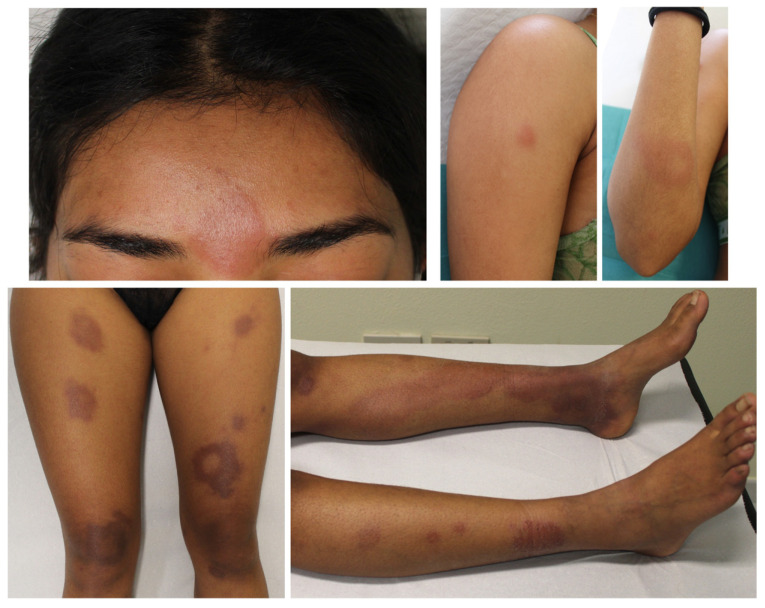
A 22-year-old patient with multiple, annular and linear, erythematous macules and papules on the upper and lower limbs and a round, erythematous-violaceous plaque on the forehead.

**Figure 2 pathogens-12-00165-f002:**
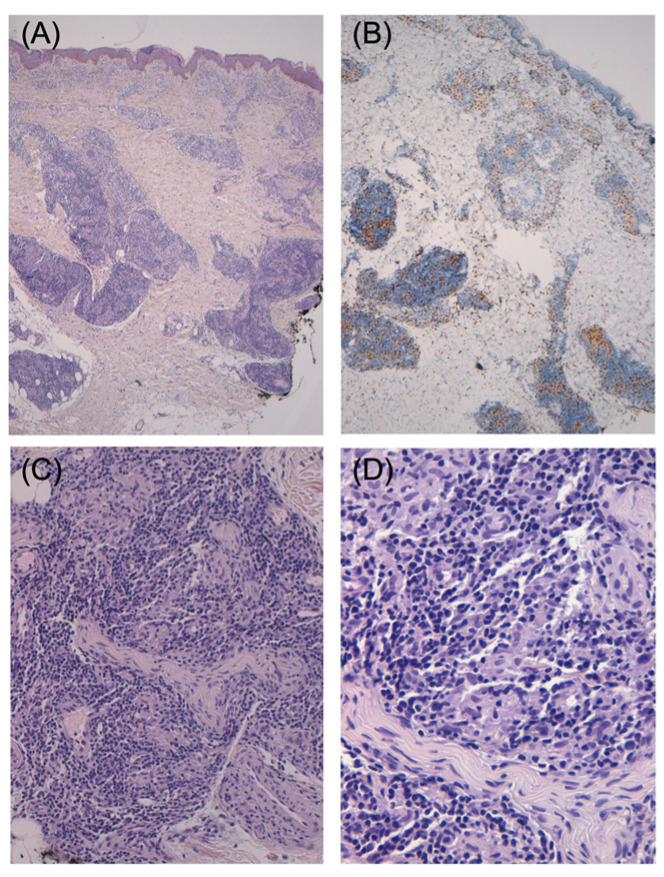
Non-necrotizing epithelioid granulomas, with a peripheral rim of lymphocytes and plasma cells, are arranged around neurovascular bundles in the deep dermis and subcutis. Epidermis and upper papillary dermis are not involved. (**A**). Hematoxylin-Eosin (final magnification: 40×): non-necrotizing epithelioid granulomas. (**B**). Immunohistochemical stain CD68/PGM1 Eosin (final magnification: 40×): epithelioid histiocytes; (**C**). Hematoxylin-Eosin (final magnification: 200×): nerve branches involved; (**D**). Hematoxylin-Eosin (final magnification: 400×): nerve branches involved and epithelioid histiocytes.

**Table 1 pathogens-12-00165-t001:** List of samples and RLEP qPCR results from both sites (Italy and USA).

Sample Name	Type of Sample	Sampling Date (DD.MM.YY)	qPCR Ct ($\bar{x}$)
3654-2136/21	Skin biopsy (right forearm)	19.04.21	**24.67**
3654-5546/21	Skin biopsy (right arm)	15.07.21	**28.52**

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
