# Peer review of "Leprosy in an Adopted Woman Diagnosed by Molecular Tools: A Case Report from a Non-Endemic Area"

_pathogens, 2023, doi:10.3390/pathogens12020165_

Round 1

Reviewer 1 Report

The authors report a case of leprosy in a 22-year-old Indian woman, adopted at the age of 10 and living in Italy. The diagnosis was confirmed in an Italian tropical diseases center (IRCCS Sacro Cuore Don Calabria Hospital, Negrar di Valpolicella, Italy).

Acid Fast Bacilli (AFB) were evidenced on histological samples, M. leprae antibodies against the specific phenolic glycolipid-I (PGL-I) were positive on serum; microscopic analysis of a nasal swab and multiple lesions were negative for AFB; the search for genomic sequences of M. leprae in real-time PCR in DNA extracted from skin biopsies confirmed the presence of M. leprae. The PCR was performed double-blind in Italy and USA.

The case is well described, very interesting for healthcare workers in non-endemic areas, and useful to raise awareness on leprosy.

Minor observations:

Page 2, line 58: AFS is probably AFB. If so it should be “acid-fast bacilli (AFB)”;  line 64 should then be only AFB and not “acid-fast bacilli (AFB)”

Page 2, line 67: SSS should be completed

Page 4, table 1:”n.d.: not detected” should be removed

Page 5, line 209: Z-N stain should be ZN

The Funding part should be revised

Author Response

Page 2, line 58: AFS is probably AFB. If so it should be “acid-fast bacilli (AFB)”;  line 64 should then be only AFB and not “acid-fast bacilli (AFB)”

Authors’ response: We apologize for the mistake. Thank you for checking the manuscript with great skill and therefore intercepting the error. We modified it as suggested.

Page 2, line 67: SSS should be completed

Authors’ response: the acronym SSS stands for slit-skin smear, and it is defined on page 2, line 65

Page 4, table 1:”n.d.: not detected” should be removed

Authors’ response: Done.

Page 5, line 209: Z-N stain should be ZN

Authors’ response: Done.

Reviewer 2 Report

Manuscript ID: pathogens-2140174
Type of manuscript: Case Report
Title: Leprosy in an adopted woman diagnosed by molecular tools: a case
report from non-endemic area
Authors: Anna Beltrame et al.

The authors have used nucleic acid amplification (NA) techniques to confirm a diagnosis of leprosy in a young female patient with pre-existing treated myasthenia gravis. DNA from the pathogen M. leprae was successfully extracted from fixed skin biopsies after some routine tests for Z-N staining of acid-fast bacilli (AFB) were negative. In addition to confirming the presence of leprosy, the authors were able to show the strain was drug sensitive and also to obtain a main SNP genotype (1D) by further analysis and Sanger sequencing.

Successful DNA extraction from previously fixed and embedded tissue can be challenging, so both the technical and clinical achievements in this case are significant. The dermatologic and rheumatic manifestations of leprosy can be a great mimic of other conditions so a correct diagnosis is important for implementing the appropriate therapy.

MAIN TEXT

Author affiliations, lines 10-12. 

6 seems to precede 5 in the list of author affiliations.

Abstract.  The genotyping of the causative strain of M. leprae was a significant achievement and could be mentioned in the Abstract.

PCR methodology.

1. Line 127. Authors please indicate the number of paraffin-embedded sections and thickness used to extract DNA with the MagCore kit.

2. Line 136. Was the BSA used acetylated or non-acetylated? If the latter, this would assist in overcoming any PCR inhibition carried over from the extraction procedure. Additionally, acetylated BSA has been associated with differential PCR inhibition. Ramalingam et al. Biomicrofluidics. 2017, 11(3):034110.

Table 1

1. Are the qPCR Ct values given the mean of duplicates or single estimations?  The methodology implies they should be duplicate estimations (p4, line 142). If so they should be identified as such here (x̄).

2. In the caption, to what test does the nd: not detected refer? This term does not appear in the Table itself.

Figure 2.

Authors please clarify the final magnification of the 4 panels shown. At present, no magnification is provided for panel B. Moreover, it looks as if the factors given are for the objective lens only (4x, 20x and 40x). The eyepiece is usually 10x, so e.g. A would be 40x final.

English grammar (minor points).

1. As the Ridley-Jopling classification scale and also the PGL-1 tests are not male, even if developed by men, I suggest author substitute “its” in place of “his” on p2, line 59 and also P6, line 226.

2. Line 162, p5.   Suggest “exhibited” or “displayed” a genotype 1D rather than  “harboured”.

References.

The following references need minor corrections for consistency.

Refs. 5, 16 and 40 are incomplete (page nos).

SUPPLEMENTARY INFORMATION.

It is good to see the convincing sequencing data presented here in its entirety. As this section may be read independently of the main text, I suggest the authors consider repeating here that the sample type for 3654-2136/21 was a skin biopsy and also add in the relevant reference  (18) for the genotyping method for SNP in ML2535.

Author Response

Author affiliations, lines 10-12. 

6 seems to precede 5 in the list of author affiliations.

 Authors’ response: We apologize for the mistake. Thank you for checking the manuscript with great skill and therefore intercepting the error. We modified it as suggested.

Abstract.  The genotyping of the causative strain of M. leprae was a significant achievement and could be mentioned in the Abstract.

 Authors’ response: Thank you for the suggestion. This information has been included in the in abstract as suggested “Genotype analysis allowed the identification of a genotype 1D of M. leprae, facilitating the epidemiological investigation of the plausible infection origin”.

PCR methodology.

  1. Line 127. Authors please indicate the number of paraffin-embedded sections and thickness used to extract DNA with the MagCore kit.

 Authors’ response: We sliced multiple 5-micron sections (n= 10 sections) per each sample. We now added the missing information.

  1. Line 136. Was the BSA used acetylated or non-acetylated? If the latter, this would assist in overcoming any PCR inhibition carried over from the extraction procedure. Additionally, acetylated BSA has been associated with differential PCR inhibition. Ramalingam et al. Biomicrofluidics. 2017, 11(3):034110.

 Authors’ response: Yes, we used non-acetylated BSA. We added this in the main text.

Table 1

  1. Are the qPCR Ct values given the mean of duplicates or single estimations?  The methodology implies they should be duplicate estimations (p4, line 142). If so they should be identified as such here (x̄).

Authors’ response: The qPCR Ct values are expressed as the mean. We now modified as suggested.

  1. In the caption, to what test does the nd: not detected refer? This term does not appear in the Table itself.

 Authors’ response: We apologize for the mistake. We deleted it.

Figure 2.

Authors please clarify the final magnification of the 4 panels shown. At present, no magnification is provided for panel B. Moreover, it looks as if the factors given are for the objective lens only (4x, 20x and 40x). The eyepiece is usually 10x, so e.g. A would be 40x final.

Authors’ response: thank you for the suggestions. We add the final magnification also for panel B

English grammar (minor points).

  1. As the Ridley-Jopling classification scale and also the PGL-1 tests are not male, even if developed by men, I suggest author substitute “its” in place of “his” on p2, line 59 and also P6, line 226.

 Authors’ response: Done.

  1. Line 162, p5.   Suggest “exhibited” or “displayed” a genotype 1D rather than  “harboured”.

 Authors’ response: Done.

References.

The following references need minor corrections for consistency.

Refs. 5, 16 and 40 are incomplete (page nos).

  Authors’ response: Done.

SUPPLEMENTARY INFORMATION.

It is good to see the convincing sequencing data presented here in its entirety. As this section may be read independently of the main text, I suggest the authors consider repeating here that the sample type for 3654-2136/21 was a skin biopsy and also add in the relevant reference  (18) for the genotyping method for SNP in ML2535.

 Authors’ response: Done.